# The Expression of Toll-like Receptors in Cartilage Endplate Cells: A Role of Toll-like Receptor 2 in Pro-Inflammatory and Pro-Catabolic Gene Expression

**DOI:** 10.3390/cells13171402

**Published:** 2024-08-23

**Authors:** Tamara Mengis, Laura Bernhard, Andrea Nüesch, Irina Heggli, Nick Herger, Jan Devan, Roy Marcus, Christoph J. Laux, Florian Brunner, Mazda Farshad, Oliver Distler, Christine L. Le Maitre, Stefan Dudli

**Affiliations:** 1Center of Experimental Rheumatology, Department of Rheumatology, University Hospital, University of Zurich, 8008 Zurich, Switzerlandirina.heggli@mssm.edu (I.H.); jan.devan@usz.ch (J.D.); oliver.distler@usz.ch (O.D.); stefan.dudli@usz.ch (S.D.); 2Department of Physical Medicine and Rheumatology, Balgrist University Hospital, University of Zurich, 8008 Zurich, Switzerland; florian.brunner@balgrist.ch; 3School of Medicine and Population Health, University of Sheffield, Sheffield S10 2RX, UK; a.nuesch@sheffield.ac.uk (A.N.); c.lemaitre@sheffield.ac.uk (C.L.L.M.); 4Leni and Peter W. May Department of Orthopaedics, Icahn School of Medicine at Mount Sinai New York, New York, NY 10029, USA; 5Department of Radiology, Balgrist University Hospital, University of Zurich, 8008 Zurich, Switzerland; 6University Spine Center Zürich, Balgrist University Hospital, University of Zurich, 8008 Zurich, Switzerland; christoph.laux@balgrist.ch (C.J.L.); mazda.farshad@balgrist.ch (M.F.)

**Keywords:** toll-like receptors, cartilage endplate cells, cartilage endplate, disc degeneration, Modic changes

## Abstract

Introduction: The vertebral cartilage endplate (CEP), crucial for intervertebral disc health, is prone to degeneration linked to chronic low back pain, disc degeneration, and Modic changes (MC). While it is known that disc cells express toll-like receptors (TLRs) that recognize pathogen- and damage-associated molecular patterns (PAMPs and DAMPs), it is unclear if CEP cells (CEPCs) share this trait. The CEP has a higher cell density than the disc, making CEPCs an important contributor. This study aimed to identify TLRs on CEPCs and their role in pro-inflammatory and catabolic gene expression. Methods: Gene expression of TLR1–10 was measured in human CEPs and expanded CEPCs using quantitative polymerase chain reaction. Additionally, surface TLR expression was measured in CEPs grouped into non-MC and MC. CEPCs were stimulated with tumor necrosis factor alpha, interleukin 1 beta, small-molecule TLR agonists, or the 30 kDa N-terminal fibronectin fragment. TLR2 signaling was inhibited with TL2-C29, and TLR2 protein expression was measured with flow cytometry. Results: Ex vivo analysis found all 10 TLRs expressed, while cultured CEPCs lost TLR8 and TLR9 expression. TLR2 expression was significantly increased in MC1 CEPCs, and its expression increased significantly after pro-inflammatory stimulation. Stimulation of the TLR2/6 heterodimer upregulated TLR2 protein expression. The TLR2/1 and TLR2/6 ligands upregulated pro-inflammatory genes and matrix metalloproteases (MMP1, MMP3, and MMP13), and TLR2 inhibition inhibited their upregulation. Endplate resorptive capacity of TLR2 activation was confirmed in a CEP explant model. Conclusions: The expression of TLR1–10 in CEPCs suggests that the CEP is susceptible to PAMP and DAMP stimulation. Enhanced TLR2 expression in MC1, and generally in CEPCs under inflammatory conditions, has pro-inflammatory and pro-catabolic effects, suggesting a potential role in disc degeneration and MC.

## 1. Introduction

The vertebral cartilage endplate (CEP) is a thin hyaline cartilage structure separating the intervertebral disc from the vertebra. Its intactness is critical for the health of the disc [1,2]. CEP degeneration has been associated with chronic low back pain (CLBP) [3], disc degeneration (DD) [4,5,6,7], and Modic changes (MC), which are vertebral bone marrow lesions adjacent to degenerated discs [8,9]. Yet, the mechanisms linking CEP degeneration to DD and MC remain unclear. In vivo and animal disc explant models showed that structural damage of the CEP can induce features of DD and MC [10]. Biochemical changes in the CEP, like dehydration, calcification, and fibrosis diminish its nutrient transport properties, leading to reduced nutrient availability in the disc, disc cell death, and DD [2,11,12]. While it has become clear that structural and biochemical changes in the CEP are important in DD and MC, the role of biological changes in the CEP have remained largely unexplored. In the intervertebral disc, the biological degenerative mechanisms are well studied, but in the CEP, which has a cell density approximately five times higher in the central CEP than in the nucleus pulposus, seven times higher in the anterior CEP than in the anterior annulus fibrosus, and about four times higher in the posterior CEP than in the posterior annulus fibrosus [13], these changes are poorly understood. Expression of matrix metalloproteinases (MMPs) and interleukins in cartilage endplate cells (CEPCs) are suggested to play a role in CEP degradation [14].

In the intervertebral disc, the expression of matrix proteases and pro-inflammatory cytokines is increased during degeneration [15,16], and the expression of pro-inflammatory cytokines is even higher with adjacent MC [17,18,19]. Together, this leads to the resorption of the disc matrix, dehydration of the disc, and disc collapse. Disc cells also express various toll-like receptors (TLRs), with TLR2 being the only one that is responsive to inflammatory milieu such as interleukin-1β exposure. Additionally, TLR2 possesses the highest number of ligands, attributed to its capability to form diverse heterodimers [20]. The presence of TLRs gives them the capacity to sense and respond to danger signals from damage-associated molecular patterns (DAMPs) and from pathogen-associated molecular patterns (PAMPs) [21]. For example, the 30 kDa N-terminal fibronectin fragment (FNf30 kDa), short hyaluronic acid fragments, and decorin are extracellular-matrix-derived DAMPs that cause inflammatory and catabolic changes in disc cells by signaling through TLR2 and TLR4 [22,23,24]. This is important, because these fragments can be generated during DD and hence trigger a vicious inflammatory–catabolic loop within the disc [25,26,27]. Growing evidence also attest a role of intradiscal bacteria, mainly *Cutibacterium acnes* (*C. acnes*) in DD and MC [28,29,30,31,32,33]. Disc cells sense *C. acnes* through TLR2 [34,35], leading to inflammatory and catabolic changes in the disc [32,36], which extend to the endplate and eventually cause marrow changes visible as MC [32]. Coregulation of the expression of the TLR/MyD88/NFκB pathway in the disc and the bone marrow at levels with MC further support a role of TLRs in MC [19]. Taken together, these studies demonstrate a strong involvement of TLRs, and in particular of TLR2, in DD and MC, and hence targeting TLR2 is a discussed treatment for DD [37,38].

In the CEP, understanding the role of TLRs, in particular of TLR2, is important because the higher cell density could cause strong local pro-inflammatory and catabolic changes. This could have detrimental consequences, because the function and stability of the thin CEP layer could quickly be impaired contributing to DD, MC, and ultimately LBP. Furthermore, sensing DAMPs and PAMPs from the disc could propel the catabolism from the disc to the bone marrow and assist in triggering MC. However, the expression and regulation of TLRs in CEPCs is unknown to date. First, we sought to determine the expression levels of TLRs in CEPCs. Second, we aimed to investigate whether there are differences in TLR expression between cartilage endplates adjacent to degenerated discs with and without Modic changes. Third, we tested whether inflammatory environments and direct antagonists of surface TLRs affect TLR expression and activate inflammatory pathways. And fourth, we aimed to determine if TLR activation can induce degradation of the cartilage endplate.

## 2. Materials and Methods

### 2.1. CEP Collection

CEPs were collected from spinal fusion surgery patients that signed informed consent for further use of surgically removed biological material. This study was approved by the local Ethics Commission #2018-01486. Inclusion criteria for the selection of the patients were the absence of current or chronic systemic inflammatory or infectious diseases, cancer, as well as no prior lumbar fusion. The CEPs were collected in sterile tubes and processed within 1.5 h after removal.

### 2.2. RNA Extraction from Tissue

A snap-frozen CEP four-millimeter biopsy was cryomilled in liquid nitrogen to extract RNA directly from CEP tissue. The resulting homogenized tissue was suspended in 350 µL of RLT buffer containing 1% β-Mercaptoethanol (Gibco, Reinach, Switzerland). Cell debris was removed by centrifuging at 500× *g* for 10 min followed by RNA isolation using RNeasy Mini Kit (QIAGEN, Hilden, Germany) according to the manufacturer’s instructions, including the optional DNAse digestion step.

### 2.3. CEPC Isolation and Culture

CEPs were enzymatically digested overnight with 0.05% collagenase P (Roche, Basel, Switzerland) in Dulbecco’s modified eagle’s medium (DMEM) (Gibco, Reinach, Switzerland) supplemented with 10% fetal calf serum (FCS) (Gibco, 10270-106), 50 U/mL penicillin streptomycin, 10 mM HEPES, and 2 mM L-Glutamine and expanded to passage 1–2 at standard cell culture conditions (37° C with 5% CO_2_). Flow cytometry analysis confirmed that the cell population consisted of CEPCs and was free from monocytic cell contamination, showing CD34^−^, CD45^−^, and CD90^+^ expression (Appendix A) [39].

### 2.4. Culture of THP1 and HEK 293T Cells

Human embryonic kidney (HEK) 293T lenti-x cells (Tahara Cellosaurus) served as a negative control for TLR expression [21] and were cultured in DMEM with 10% FCS, 50 U/mL penicillin streptomycin, 2 mM L-Glutamine, and 6 g/L glucose prior to use. THP1 blue NF-κB SEAP reporter cells (human leukemia monocytic cell line with NF-κB SEAP Reporter) (Invivogen, San Diego, CA, USA) served as a positive control for TLR expression and were cultured according to the distributor’s instructions in RPMI 1640 supplemented with 2 mM L-glutamine, 10 mM HEPES, 10% heat-inactivated FCS, Pen-Strep 100 U/mL, 1 mM sodium pyruvate, 4.5 g/L glucose (Gibco), 100 µg/mL Normocin (Invivogen), 200 µg/mL Zeocin (Invivogen), and 200 µg/mL G418 (Invivogen).

### 2.5. Immunohistochemistry

Immunohistochemistry (IHC) was performed as previously described [40]. Human IVD tissue was obtained from 6 patients undergoing spinal surgery for nerve root pain (Sheffield Research Ethics Committee (IRAS: 10266). The tissue was fixed in 10% (*w*/*v*) formalin (Leica, Milton Keynes, UK) embedded into paraffin wax, and 4 µm sections were prepared using a microtome and mounted on positively charged slides. The sections were de-waxed and rehydrated, and the endogenous peroxidases were blocked prior to antigen retrieval. Antigen retrieval for TLR2 consisted of enzyme antigen retrieval [0.1% *w*/*v* α-chymotrypsin (Sigma Aldrich, Poole, UK) in TBS (20 mmol/L Tris, 150 mmol/L NaCl pH7.5) containing 0.1% *w*/*v* CaCl_2_ for 30 min at 37 °C], whilst for TLR4, heat antigen retrieval was used [0.05 M Tris, pH 9.5 preheated to 60 °C prior to 5 min within a rice steamer]. Following washing in TBS, nonspecific antibody binding was blocked for 1 h at room temperature in 1% (*w*/*v*) bovine serum albumin (BSA) containing 25% (*v*/*v*) goat serum (Sigma) for TLR2 or 25% (*v*/*v*) rabbit serum (Sigma) for TLR4 within tris-buffered saline (TBS). Primary antibodies were applied to the slides overnight at 4 degrees at a dilution of 1:500 for TLR2 (ab226913, Abcam, Cambridge, UK) and 1:100 dilution for TLR4 (ab22048, Abcam) in TBS containing 1% (*w*/*v*) BSA. Simultaneously, IgG controls in place of the primary antibodies were used at equal protein concentrations to test for nonspecific binding of the isotype. Following the overnight incubation, sections were washed three times in TBS, prior to the application of secondary antibodies (TLR2: goat anti-rabbit IgG (1:500, ab6720, Abcam); TLR4: rabbit anti-mouse IgG (1:400, ab6727, Abcam)) for 30 min at room temperature. Following 3 TBS washes, Elite^®^ABC reagent (Vector, Laboratories, Peterborough, UK) was added to the slides for 30 min at room temperature. After another 3 TBS washes, 0.65 mg/mL 3,3′-diaminobenzides tetrahydrochloride (Sigma-Aldrich) containing 0.08% (*v*/*v*) H_2_O_2_ in TBS was added for 20 min, prior to washes in running tap water for 5 min. The nuclei were counterstained with Haematoxylin for 20 s and blued for 3 min under running tap water. After dehydration in graded ethanol and clearing in xylene, the slides were mounted using Pertex^®^ (Leica). The slides were scanned at 20× magnification using a slidescanner (PANNORAMIC^®^ 250 Flash II DX, 3DHistech (Budapest, Hungary)), and representative images were included to highlight immunohistochemical staining.

### 2.6. Stimulation of Cultured CEPCs

Prior to stimulation, all cells were treated with the endotoxin scavenger polymyxin B (10 ng/mL) (Invivogen, Toulouse, France) for two hours. Cells were seeded in 6-well plates at a density of 10,000 cells/cm^2^. To simulate the TLR-signaling-independent inflammatory milieu provided by the degenerating disc, CEPCs were treated for 24 h or 48 h with 10 ng/mL tumor necrosis factor α (TNF-α) (Pepro Tech, London, UK) or 10 ng/mL interleukin 1 beta (IL-1β) (Pepro Tech). To selectively activate the TLR2/1, TLR2/6, and TLR4, which are the most important receptors for extracellular-matrix-derived DAMPs or bacterial-cell-wall-derived PAMPs [41], selective ligands were administered to the cell cultures for 24 and 48 h. For TLR2/6- and TLR2/1-specific activation, the synthetic ligands Pam2CysSerLys4 (Pam2csk4) and Pam3CysSerLys4 (Pam3csk4) (both Invivogen) were added, at 10 ng/mL and 1000 ng/mL, respectively. TLR4-specific activation was targeted with ultrapure E. coli lipopolysaccharide (LPS) (Invivogen) at 10 µg/mL and 50 µg/mL. In addition, the 30 kDa N-terminal fibronectin fragment (FNf30 kDa) (Sigma-Aldrich, Buchs, Switzerland) (2.5 µg/mL and 5 µg/mL) was used to represent a potential activation through extracellular-matrix-derived DAMPs.

To test for the specificity of Pam2csk4 signaling through TLR2, CEPCs were pre-treated for 2 h with 200 µM, 100 µM and 50 µM of the TLR2 inhibitor TL2-C29 (Invivogen) before 10 ng/mL Pam2csk4 was added. TL2-C29 blocks both TLR2/1 and TLR2/6 heterodimer signaling.

### 2.7. RNA Isolation from Cells

Cell lysis was conducted in RLT buffer containing 1% β-Mercaptoethanol (Gibco), and RNA isolation was carried out using the RNeasy Mini Kit (QIAGEN) following the manufacturer’s instructions, including the optional DNAse digestion step.

### 2.8. Gene Expression Analysis

Reverse transcription of 100 ng RNA was performed using the SensiFAST cDNA Synthesis Kit (Meridian Bioscience, Cincinnati, OH, USA). Relative mRNA levels were quantified with use of the SensiFAST SYBR No-Rox kit (Labgene, Châtel-Saint-Denis, Switzerland) on a magnetic induction real-time qPCR cycler (Labgene). The cycle conditions after initial denaturation at 95 °C for 300 s were 40 cycles of 5 s at 95 °C, 20 s at 60 °C, and 10 s at 72 °C, followed by melting curve analysis. Analysis was conducted with the ΔΔC_T_ method and with normalization to the reference gene Glyceraldehyde-3-Phosphate Dehydrogenase (GAPDH). All used primer sequences are listed in Table 1.

Basal TLR expression was measured in unstimulated CEPCs of 23 discs, THP1 cells, and HEK 293T cells. Additionally, unstimulated CEPCs were divided into two groups: non-MC and MC1 CEPCs, based on the presence of MC1 in the adjacent bone marrow. In stimulation experiments, a total of 11 discs were used. The expression of TLR1, TLR2, TLR4, and TLR6 of the pro-inflammatory genes IL-6, IL-8, and CCL2 and of the matrix metalloproteases MMP1, MMP3, MMP9, and MMP13 was measured in stimulation experiments. For TLR2 inhibition experiments, a total of 6 discs were used, and IL-6, TLR2, MMP1, MMP3, and MMP13 gene expression was measured.

### 2.9. Flow Cytometry

Flow cytometry was utilized to assess the surface expression levels of TLR-2 (Biolegend, San Diego, CA, USA) in unstimulated CEPCs, 72 h of 10 ng/mL, and 1000 ng/mL Pam2csk4-stimulated CEPCs, as well as 72 h of 10 ng/mL and 1000 ng/mL Pam3csk4-stimulated CEPCs. Cultured HEK293T cells were used as a negative control, and THP1 NF-κB reporter cells were used as a positive control. The cells were blocked according to the manufacturer’s instructions with 5 µL of True-Stain Monocyte Blocker™ for 10 min and stained for 45 min with 10 µg/mL monoclonal PE anti-human CD282 (TLR2) antibody (isotype: Mouse IgG2a, κ) (Biolegend). After washing, the cells were stained for 10 min with 4′,6-Diamidino-2-Phenylindole, Dilactate (DAPI) to exclude dead cells. The washed cells were analyzed on a BD LSRFortessa (BD Biosciences, Franklin Lakes, NJ, USA). The data were analyzed with FlowJo v10.8 software. Cell doublets and dead cells were excluded from the analysis.

### 2.10. CEP Explant Model

To investigate whether TLR2 stimulation in CEP explants causes features characteristics of CEP degeneration, CEP explants were stimulated with 10 µg/mL Pam2csk4. Fresh CEP was washed in PBS, and any attached disc tissue was removed. A 4 mm punch biopsy was taken and cut in half. Both halves were incubated at standard cell culture conditions in Dulbecco’s modified eagle’s medium F-12, no glutamine (DMEM) (Gibco, Reinach, Switzerland) supplemented with 10% fetal calf serum (FCS), 50 U/mL penicillin streptomycin, 10 mM HEPES, and 2 mM L-Glutamine with only one half exposed to 10 µg/mL Pam2csk4. The samples were used to assess degeneration by quantifying the release of GAGs to the media using the 1,9-dimethylmethylene blue (DMMB) assay. Briefly, both the tissue and the supernatant were gathered, and the tissue was lyophilized. Subsequently, the tissue was digested overnight in 1 mL solution of 1 mg/mL papain from papaya latex (Sigma). To eliminate debris, a centrifugation step of 10 min at 10,000× *g* was performed before carrying out the DMMB assay, which enables the detection of sulfated glycosaminoglycans (GAGs), on both the supernatant and the digested tissue mixture [42]. Chondroitin sulfate sodium salt from shark cartilage (Sigma-Aldrich, Darmstadt, Germany) was used to create the standard curve. The combination of the supernatant and the tissue mixture accounted for 100% of the GAGs initially found in the tissue. This allowed us to calculate of the proportion of GAGs released into the supernatant during the 14-day incubation period.

To evaluate gene expression in the CEP biopsy after 14 days of tissue culture, the tissue was collected and pulverized by cryomilling. The powder was immediately dissolved in RLT lysis buffer with 1% beta-mercaoptoethanol. Subsequently, RNA extraction, cDNA synthesis, and qPCR were carried out using the above-described protocol.

### 2.11. Statistical Analysis

To assess the relationship between age and the expression levels of surface TLRs, the Pearson correlation was used, and statistical significance was determined using two-tailed *p*-values.

The 2^-DC_T_ values of TLRs measured in CEP tissue were compared to the 2^-DC_T_ values measured in HEK and THP1 cells using one-way ANOVA, followed by Dunnett’s multiple comparisons testing. For comparison of TLR expression of non-MC to MC1 CEPCs, the DC_T_ values were normalized to the average DC_T_ value of non-MC CEPCs. Statistical analysis was then performed on the resulting DDC_T_ values using an unpaired *t*-test. Statistical analysis of all qPCR data from stimulation and inhibition experiments was conducted by one-way analysis of variance (ANOVA) on the log_2_ fold change in ΔΔC_T_ values normalized to unstimulated CEPCs, followed by Dunnett’s multiple comparisons test. Correlation of TLR2 to TLR1 or TLR6 was tested by fitting a simple linear regression model. For flow cytometry experiments, differences in median fluorescence intensity were tested with one-way ANOVA followed by Dunnett’s multiple comparisons test. The percentage of GAGs released into the medium using the CEP explant model was analyzed by a one-tailed Wilcoxon test. In addition, a binominal test was used to determine if the proportion of CEP biopsy punch pairs which found a greater release in the stimulated condition than in the unstimulated condition reaches significance.

All statistical analyses were performed using GraphPad Prism V10.2.0. The significance level was α = 0.05, if not stated otherwise.

## 3. Results

### 3.1. Patient Demographics

Different patient groups were utilized for the experiments performed because of the limited number of cells isolated from each patient and the low passage number used.

Patients from which CEPs for ex vivo TLR expression analysis were taken had a mean age of 64, a mean BMI of 26.6, with 33.3% being female. The disc levels were L4/5 and L5/S1 with a median Pfirrmann grade of 5 (Appendix A). IHC was performed on the endplates of six different female patients, ranging in age from 43 to 70 years. CEP of both non-MC and MC patients were included (Appendix A).

In patients used for the analysis of TLR1, 2, 4, and 6 expressions on CEPCs isolated from MC1 (*n* = 6) and non-MC (*n* = 10) CEPs, there were no significant differences in demographics including age, BMI, weight, and height (Appendix A). The Pfirrmann grade of the discs adjacent to the CEP also did not differ between the groups with a median Pfirrmann grade of 4 (interquartile range (IQR) = [3.75, 5.00]) in the non-MC group and 4.5 (IQR = [3.00, 5.00]) in the MC1 group (*p* = 0.68).

The CEPs used for the stimulation and inhibition experiments originated from the lumbar levels L3/4, L4/5, and L5/S1, and the corresponding discs had an average Pfirrmann grade of 3.8 (range 2–5). The age of the patients included ranged from 28 to 87 years. Both men and women were included (Appendix A).

No correlation was identified between age and the expression levels of surface TLRs, with weak and non-significant associations observed for TLR1 (r = 0.259, *p* = 0.32), TLR2 (r = −0.360, *p* = 0.16), TLR4 (r = 0.255, *p* = 0.32), and TLR6 (r = 0.159, *p* = 0.54).

### 3.2. TLR Expression in CEP Tissue

Gene expression of all TLRs (TLR1-10) was detected in ex vivo CEP tissue (*n* = 3). The expression of all TLRs was significantly higher than in HEK cells, which were used as a negative control for TLR expression. Additionally, expression levels of TLR1, 3, 4, 9, and 10 in CEP tissue were significantly higher compared to the positive-control THP1 cells (Figure 1A,C,D,I,J). The other TLRs (TLR2, 5, 6, 7, and 8) had similar expression levels to THP1 cells (Figure 1A–J).

The expression of TLR2 and TLR4, known for recognizing DAMPs and PAMPs, was further examined using IHC. TLR2 and TLR4 show similar expression patterns, being present mainly in the CEPCs near the intervertebral disc, while cells adjacent to the vertebral bone showed no TLR2 and TLR4 expression (Figure 1K,L).

### 3.3. Basal TLR Expression in Cultured CEPCs

TLR8 and TLR9 expression was lost in isolated and expanded CEPCs after one passage whereas gene expression of all other TLRs could be detected (Appendix A). Since TLR2 can dimerize with TLR1 and TLR6, correlation of TLR2 with these two TLRs was tested (Appendix A). TLR2 expression only correlated with TLR6 expression (R = 0.581, *p* = 0.004).

### 3.4. TLR2 Significantly Increased in MC1 CEPCs

After categorizing CEPCs into MC1 CEPC (*n* = 6) and non-MC CEPC (*n* = 9) groups, we measured differences in surface TLRs, specifically TLR1, TLR2, TLR4, and TLR6. TLR2 and TLR4 are primarily responsible for recognizing PAMPs and DAMPs, along with TLR1 and TLR6 that dimerize with TLR2. TLR2 (*p* = 0.029) was found to be significantly upregulated in the MC1 CEPCs, while TLR1 (*p* = 0.29) and TLR4 (*p* = 0.28) did not exhibit significant differences between the two groups (Figure 2A–C). TLR6 showed a slight upregulation; however, it did not reach statistical significance (*p* = 0.07) (Figure 2D).

### 3.5. Regulation of TLR1, 2, 4, and 6 Expression under Inflammatory Conditions

TLR1 was only significantly increased upon 24 h (*p* < 0.001) and 48 h (p = 0.030) of IL-1β stimulation as well as 48 h 10 ng/mL TNF-α stimulation (*p* < 0.001) (Figure 3A). TLR2 was significantly upregulated after 24 h in response to 10 (*p* = 0.007) and 100 ng/mL (*p* < 0.001) Pam2csk4, 1000 ng/mL Pam3csk4 (*p* = 0.030), and 10 ng/mL TNF-α (*p* < 0.001) (Figure 3B). After 48 h, all the applied stimulations, including Fnf30 kDa, induced a significant increase in TLR2 expression (Figure 3B). In contrast, TLR4 expression was not changed by any of the applied stimulations (Figure 3C). For TLR6 expression, only a 48 h exposure to 10 ng/mL LPS (*p* = 0.042) and 10 ng/mL TNF-α (*p* = 0.009) induced a significant increase (Figure 2D).

### 3.6. Inflammatory Gene Upregulation upon TLR Activation

Exposure to TLR ligands led to an inflammatory response, as indicated by the heightened expression of three key inflammatory genes: IL-6, IL-8, and CCL2. All three measured genes showed a similar upregulation pattern. After 24 h, all experimental conditions induced a significant increase in response, except for FNf30 kDa and the lower dosage of 10 ng/mL Pam3csk4 (Figure 4). However, after 48 h, in addition to the other stimulations, the 5 µg/mL FNf30 kDa also resulted in a significant increase in IL-6 expression (*p* = 0.006) (Figure 4).

### 3.7. Protease Upregulation through TLR2/6 Activation

Stimulation with TLR agonists as well as inflammatory cytokines TNF-α and IL-1β upregulated several matrix proteases in CEPCs after 24 h. MMP1 and MMP3 were significantly increased through 10 ng/mL and 1000 ng/mL Pam2csk4, 1000 ng/mL Pam3csk4, 50 ng/mL LPS, 10 ng/mL TNF-α, and 10 ng/mL IL-1β (Figure 5A,B). This pattern persisted after 48 h, with the addition of FNf30 kDa, which after this time also increased MMP1 and MMP3 significantly (Figure 5A,B). For MMP9, the response pattern was similar to MMP1 and MMP3 with the difference that FNf30 kDa could not upregulate MMP9 expression (Figure 5C). After 24 h, MMP13 exhibited a less pronounced response than the other MMPs. Only Pam2csk4 (*p* = 0.037 and *p* = 0.001 for 10 ng/mL and 1000 ng/mL, respectively), IL-1β (*p* = 0.002), and TNF-α (*p* = 0.011) upregulated its expression. However, after 48 h, MMP13 showed a significant increase in response to all applied stimulations, except for the lower dosage of Pam3csk4 and the lower concentration of FNf30 kDa (Figure 5D). Among all measured proteases, MMP3 demonstrated the strongest response. A 48 h exposure to 1000 ng/mL Pam2csk4 increased expression by an average fold change in 856.3 ± 456.2 (Figure 5B).

### 3.8. Stimulation of TLR2/6 Heterodimer Increases TLR2 on CEPC Cell Surfaces

Flow cytometry showed more TLR2 on CEPCs compared to HEK cells (used as a negative control) and a lower number compared to the monocyte cell line THP1 (Figure 6A). A significant increase in surface TLR2 on CEPCs (fold change = 1.4, *p* = 0.005) was observed after a 72 h stimulation with 1000 ng/mL Pam2csk4 (Figure 6B) but not with Pam3csk4 (Figure 6C).

### 3.9. TL2-C29 Inhibits TLR2 Signaling

The TLR2 inhibitor TL2-C29 attenuated the Pam2csk4-mediated upregulation of pro-inflammatory and catabolic gene expression. The addition of 50 µM TL2-C29 was only sufficient for creating a significant decrease in IL-6 (*p* < 0.001) and MMP3 (*p* = 0.034) expression (Figure 7A,D). However, with the addition of 100 µM and 200 µM of the TL2-C29 inhibitor, the upregulation of IL-6, TLR2, MMP1, MMP3, and MMP13 induced by Pam2csk4 were also significantly reduced (Figure 7A–E).

### 3.10. TLR2 Stimulation Induces Degeneration in CEP Explant Model

The potential of TLR2 signaling for inducing CEP damage was evaluated due to its consistent upregulation of both catabolic and inflammatory cytokines as well as its own expression. The CEP explant model showed that exposure of the CEP to the synthetic TLR2 ligand Pam2csk4 resulted in a significant release of GAGs (*p* = 0.020) into the medium compared to no stimulation after 14 days (Figure 8A). The use of a binomial test to compare the proportion of biopsy punch pairs with greater GAG release in the stimulated condition than in the unstimulated condition found that significantly more pairs showed an increase than expected by chance (*p* = 0.038), with 18 out of 26 pairs showing an increased GAG release in the Pam2csk4-stimulated condition.

The direct isolation of RNA from the CEP pieces after 14 days of incubation found that of the measured MMPs (MMP1, MMP3, and MMP13), only MMP3 (*p* < 0.001) showed a significant increase in the explant model when stimulated with Pam2csk4. Furthermore, the significant upregulation of the inflammatory cytokine IL-6 (*p* = 0.010) observed in the 2D cell culture was also seen in this 3D explant model. All TLRs show a trend towards upregulation; however, only TLR2 (*p* = 0.044) and TLR4 (*p* = 0.042) achieved significant upregulation, while TLR1 (*p* = 0.10) and TLR6 (*p* = 0.10) did not reach significance, mirroring the effects in 2D cell cultures (Figure 8B).

## 4. Discussion

The main findings of this study include the demonstration that CEPCs express all TLRs except for TLR8 and TLR9; TLR2 expression was increased in MC1 CEPCs and following stimulation with inflammatory cytokines. Furthermore, TLR2 stimulation upregulates its own expression and triggered catabolic changes in CEP tissue similar to those induced by inflammatory cytokines. This suggests a potential role of TLR2 signaling in CEPs in disc degeneration and MC1.

This is the first systematic study showing that CEPCs express all the TLRs, TLR1-10. Interestingly, TLR8 and TLR9 expression was lost when the cells were cultured and expanded. Expression of the TLR2 and TLR7 genes have previously been reported using micro arrays [43]. Single-cell RNA sequencing and bulk RNA sequencing showed that CEPCs are transcriptionally different from nucleus pulposus and annulus fibrosus cells of the intervertebral disc, and hence, CEPCs should be investigated separately [43,44]. Therefore, the expression of TLR7 and TLR8 seems to be specific to CEPCs [21]. This underscores that CEPCs should be investigated separately from disc cells.

CEPCs expressed all cell surface TLRs, i.e., TLR1, TLR2, TLR4, TLR5, and TLR6. Of those, TLR2 and TLR4 are of particular interest in DD and MC, because they can be activated by extracellular matrix fragments generated during DD and by *C. acnes*, a bacterium that has been found at increased concentrations in discs at MC levels and that can trigger DD and MC in animal models [22,32,33,36,41,45]. Both TLR2 and TLR4 were identified using IHC in the human CEP, although their expression was seen to be often restricted to the internal disc region of the CEP which attaches to the nucleus pulposus. In contrast, cells near the vertebral bone exhibited limited TLR2 and TLR4 immunopositivity. This suggests that the cells within the CEP exhibit spatially different gene expression profiles, or that the cells facing the nucleus pulposus have previously been exposed to TLR2 and TLR4 ligands, resulting in the upregulation of these receptors.

TLR2 signaling requires heterodimerization with TLR1 or TLR6 [22]. TLR4 homodimerizes for activating downstream signaling. Interestingly, only TLR2 was significantly increased in MC1 compared to non-MC cultured CEPCs, with TLR6 being slightly but not significantly elevated, indicating potentially increased signaling through the TLR2/6 heterodimer in MC1.

To test the regulation of TLR1, 2, 4, and 6 under inflammatory conditions, CEPCs were stimulated with the TLR1/2- and TLR2/6-specific ligands Pam3csk4 and Pam2csk4, with TLR4-specific ultrapure LPS, and with IL-1β or TNF-α to test TLR-signaling-independent regulation [21]. TLR2 expression was found to be upregulated under all these conditions. TLR4 was not affected at all, while TLR1 was only upregulated by TNF-α and TLR6 was only upregulated by LPS and TNF-α. At the protein level, stimulation with Pam2csk4 but not with Pam3csk4 enhanced TLR2 expression, indicating a positive feedback loop of TLR2/6 stimulation with TLR2 expression. The significant correlation of TLR2 expression with TLR6 but not with TLR1 further supports the relevance of TLR2/6 signaling in CEPCs. Dissecting downstream signaling in future studies could further support the relevance of TLR2/6 signaling in CEPCs. In intervertebral disc cells and chondrocytes from osteoarthritic joints, similarly to our data, only TLR2 was upregulated after IL-1β stimulation [21,46]. Together, this suggests that TLR2/6 signaling induces a positive feedback loop, possibly indicating that the increased TLR2 expression observed in MC1 CEPCs results from previous exposure to TLR2/6 ligands in vivo. Whether this affects sensitivity or intensity of TLR2/6 signaling in vivo cannot be concluded from these experiments.

Expression of pro-inflammatory cytokines and MMPs was enhanced after TLR2 and TLR4 signaling in the disc and in hyaline cartilage, leading to matrix resorption and impaired function [21,23,24,38,46,47,48]. Here, stimulation of TLR1/2, TLR2/6, and TLR4 enhanced the expression of pro-inflammatory cytokines and of MMPs, suggesting a potential detrimental role of TLR2 and TLR4 signaling in endplate degradation [14]. Blocking TLR2 signaling abrogated the TLR2/6 enhanced expression of pro-inflammatory cytokines and MMPs. This demonstrated that signaling indeed occurred through TLR2 and that the observed upregulation of cytokines and MMPs was functionally linked to TLR2 stimulation. TLR2 stimulation in a CEP explant culture model demonstrated the catabolic effect of TLR2 signaling in CEPs, as evidenced by the reduction in GAGs. While this does not confirm the relevance of TLR2 signaling in CEP degradation in DD and MC1, it shows a potential detrimental effect of TLR2 ligands in endplate resorption, which is observed in DD and MC1 [9].

TLR2 signaling could occur in CEPCs through DAMPs or PAMPs that have been shown to be present in degenerating intervertebral discs. For example, FNf30 kDa generated during DD activates TLR2 [23,26,27], and *C. acnes* activates TLR2 [34]. However, which *C. acnes* factors activate TLR2 is unclear. In keratinocytes during acne vulgaris, Christie–Atkins–Munch-Petersen 1 factor (CAMP1) binds TLR2 [49], and di- and tri-acylated lipoproteins engage TLR1/2 and TLR2/6 heterodimers, respectively [50]. *C. acnes* also secrete hyaluronidase, which can cleave hyaluronic acid into fragments acting as DAMPs [24,51], and sialidases, which can disrupt TLR-inhibitory mechanisms [49,52]. While it can be expected that CEPCs respond to *C. acnes*, this needs to be demonstrated.

The overexpression of MMPs could also degrade the matrix of the CEP which is thin anyway and increase the risk for damage with detrimental consequences like MC1 [53], avulsion-type herniations [54], Schmorl’s nodes [55], and endplate fractures [56]. Lastly, the overexpression of pro-inflammatory cytokines and MMPs by CEPCs may also affect the adjacent bone marrow and lead to inflammatory changes even before the CEP is damaged.

This study is limited by the fact that the significance of CEPC TLR signaling for the pathogenesis of DD and MC1 cannot be derived from the experiments performed. The degeneration of the endplate observed using the CEP explant model, which included gene expression analysis and GAG release measurement after prolonged incubation, suggests a positive feedback loop involving TLR expression and a destructive mechanism following TLR2 activation. However, it does not establish a direct connection between the upregulated MMPs and the released ECM material. In addition, the downstream mechanisms of TLR activation cannot be determined from this study.

## 5. Conclusions

In conclusion, this study demonstrates that CEPCs express TLRs and that TLR signaling leads to the overexpression of inflammatory cytokines and various MMPs. Specifically, TLR2, which was found to be increased in cultured MC1 CEPCs, can initiate destructive effects on CEP tissue. As the CEP has a critical function in maintaining disc health, the CEP damage seen here, at least in part, may be regulated via TLRs which could contribute to DD and MC.

## Figures and Tables

**Figure 1 cells-13-01402-f001:**
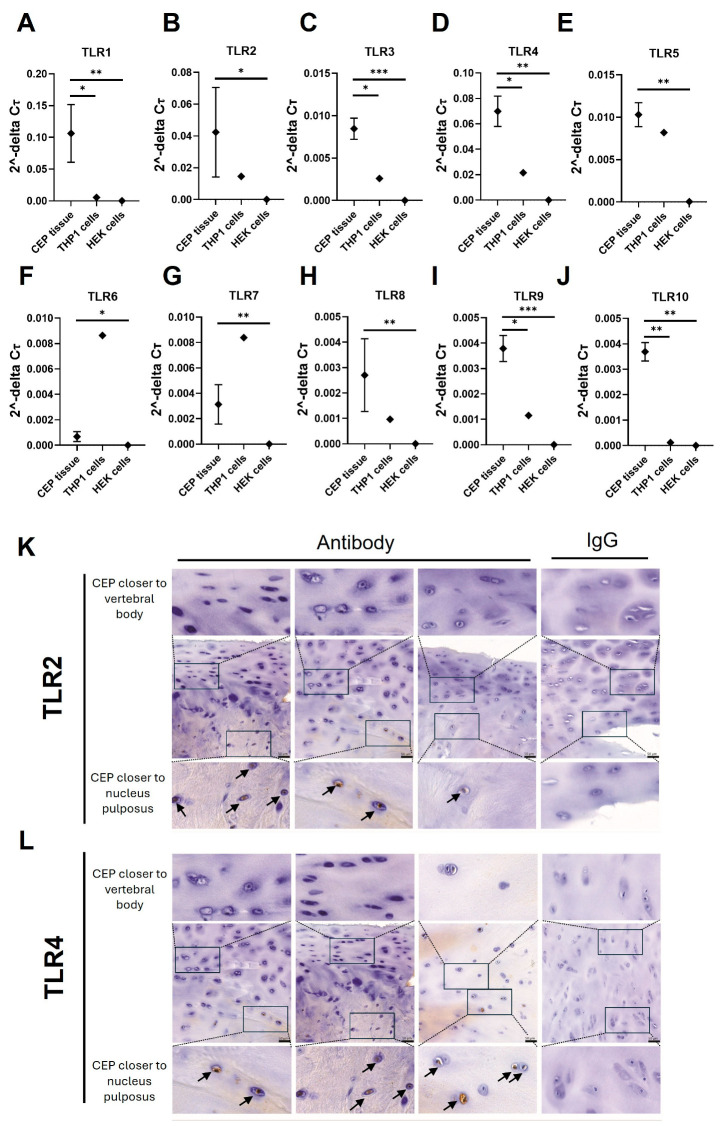
Ex vivo analysis of TLR gene expression in CEP tissue. (**A**–**J**) TLR1–10 were measured with qPCR in RNA isolated from freshly isolated CEP tissue, HEK cells, and THP1 cells. The graphs present the 2^-DC_T_ values normalized to GAPDH. Significant differences in expression levels between CEP tissue, THP1 cells, and HEK cells were tested using ordinary one-way ANOVA followed by Dunnett’s multiple comparisons test. * *p* < 0.05, ** *p* < 0.01, *** *p* < 0.001. (**K**,**L**) Immunohistochemistry stains for (**K**) TLR2 and (**L**) TLR4 from the CEP of three representative patients as well as the isotype control within a single patient example are shown (right column). Positive staining was found mainly on the side of the CEP closer to the nucleus pulposus. Positive staining is indicated by brown coloration and nuclei counterstain with Hematoxylin. Scale bars = 50 µm as indicated in the central image; zoomed-in regions are indicated with rectangles and positive stainings are highlighted by arrows.

**Figure 2 cells-13-01402-f002:**
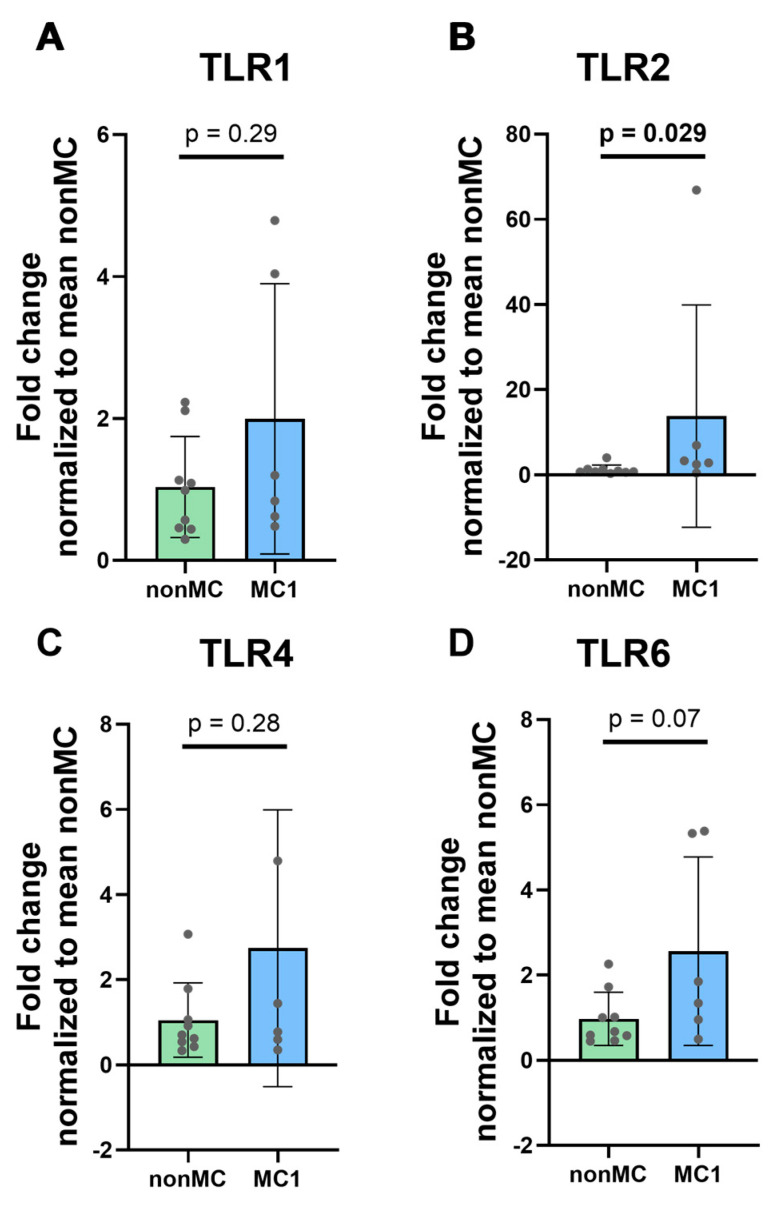
Expression of surface TLRs involved in DAMP and PAMP recognition; specifically, (**A**) TLR1, (**B**) TLR2, (**C**) TLR4, and (**D**) TLR6 were measured in unstimulated cultured CEPCs isolated from CEPs adjacent to a degenerated disc with adjacent MC1 (*n* = 6) and from CEPs adjacent to a degenerated disc without MC1 (non-MC) (*n* = 9). The graphs illustrate mean fold change ± standard deviation. Significance was tested using unpaired *t*-tests on log_2_ fold change values normalized to the mean of non-MC DCt values. The graphs indicate mean fold change values ± standard deviation. The bold *p*-values indicate significance (*p* < 0.05).

**Figure 3 cells-13-01402-f003:**
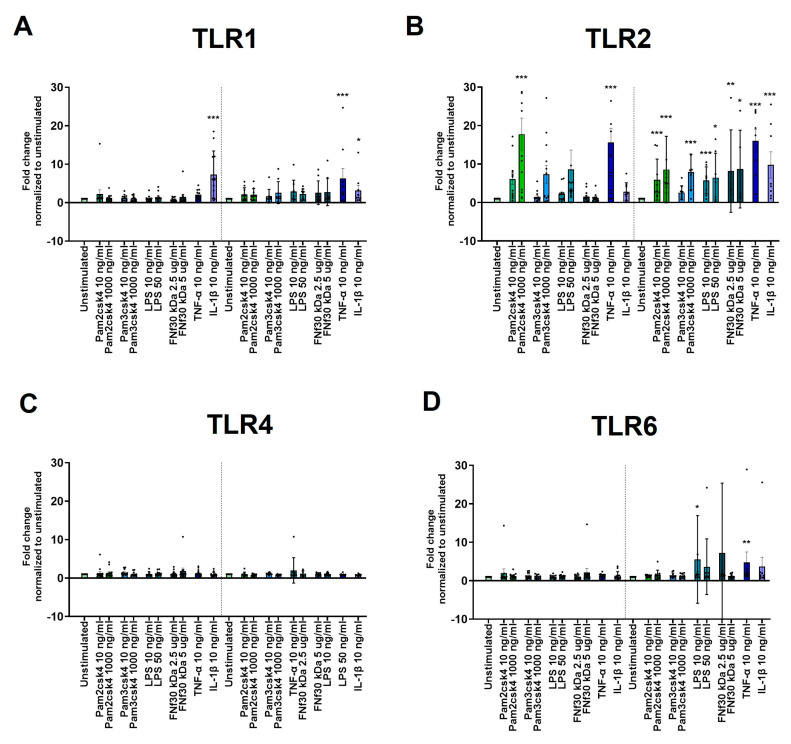
In vitro CEPC TLR expression and regulation. (**A**–**D**) Gene expression of TLRs in CEPCs following 24 h and 48 h stimulation with varying concentrations of Pam2csk4, Pam3csk4 (10 ng/mL and 1000 ng/mL), LPS (10 ng/mL and 50 ng/mL), FN fragment 30 kDA (FNf 30kDa) (2.5 µg/mL and 5 µg/mL), TNF-α (10 ng/mL), and IL-1β (10 ng/mL) is depicted. The panels illustrate the expression levels of (**A**) TLR1, (**B**) TLR2, (**C**) TLR4, and (**D**) TLR6. Significance was tested on log_2_ fold change in DDC_T_ values normalized to unstimulated CEPCs by repeated measures one-way ANOVA, followed by multiple comparisons which compared each condition to the unstimulated condition at the respective timepoint. The graphs indicate mean fold change values ± standard deviation. Dunnett’s statistical hypothesis testing was applied to correct for multiple comparisons. The asterisks signify significance: * *p* < 0.05, ** *p* < 0.001, *** *p* < 0.0001.

**Figure 4 cells-13-01402-f004:**
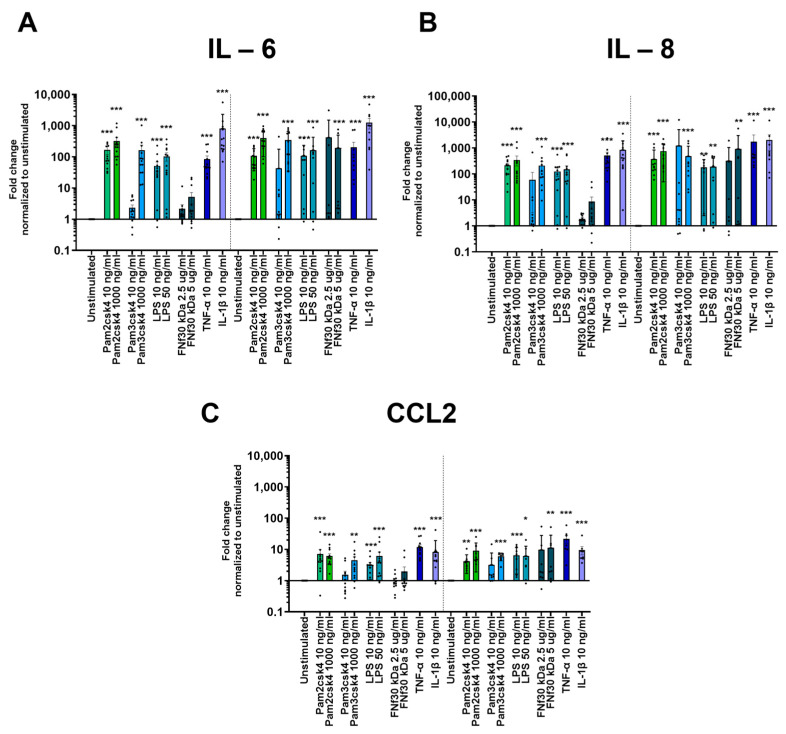
Inflammatory gene expression of (**A**) IL-6, (**B**) IL-8, and (**C**) CCL2 in CEPCs following 24 h and 48 h stimulation with varying concentrations of Pam2csk4, Pam3csk4 (10 ng/mL and 1000 ng/mL), LPS (10 ng/mL and 50 ng/mL), FN fragment 30 kDa (FNf30 kDa) (2.5 µg/mL and 5 µg/mL), TNF-α (10 ng/mL), and IL-1β (10 ng/mL) is depicted. The graphs indicate mean fold change values ± standard deviation. Significance was tested on log_2_ fold change in D D CT values by repeated measures one-way ANOVA, followed by multiple comparisons which compared each condition to the unstimulated condition at the respective timepoint. Dunnett’s statistical hypothesis testing was applied to correct for multiple comparisons. The asterisks signify significance: * *p* < 0.05, ** *p* < 0.001, *** *p* < 0.0001.

**Figure 5 cells-13-01402-f005:**
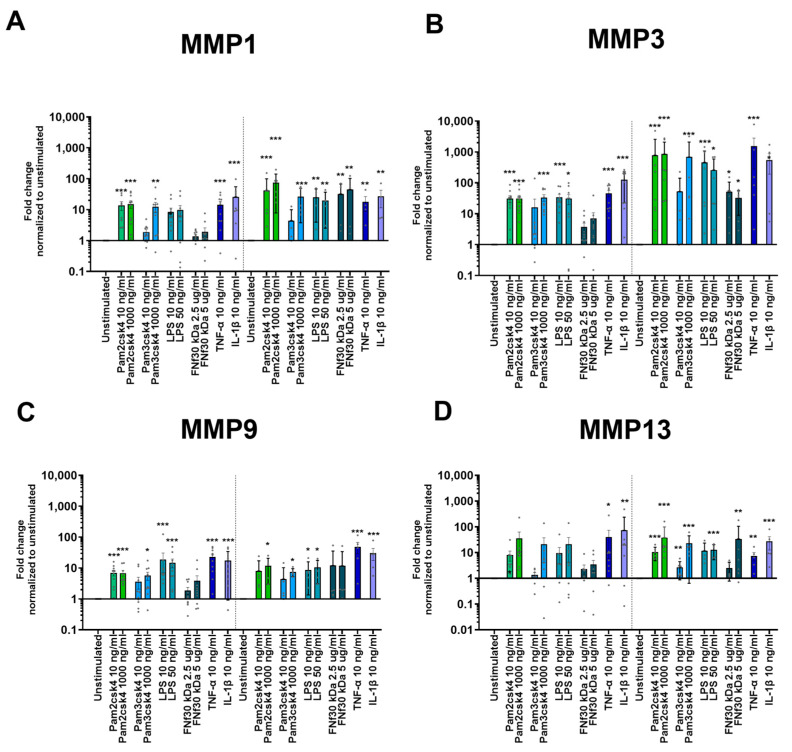
Protease gene expression of (**A**) MMP1, (**B**) MMP3, (**C**) MMP9, and (**D**) MMP13 in CEPCs after either 24 or 48 h of Pam2csk4, Pam3csk4 (10 ng/mL and 1000 ng/mL), LPS (10 ng/mL and 50 ng/mL), FN fragment 30 kDA (FNf) (2.5 µg/mL and 5 µg/mL), TNF-α (10 ng/mL), and IL-1β (10 ng/mL) stimulation. The graphs indicate mean fold change values ± standard deviation. Significance was tested on log2 fold change in D D CT values normalized to unstimulated CEPCs by repeated measures one-way ANOVA, followed by multiple comparisons which compared each condition to the unstimulated condition at the respective timepoint. Dunnett’s statistical hypothesis testing was applied to correct for multiple comparisons. The asterisks signify significance: * *p* < 0.05, ** *p* < 0.001, *** *p* < 0.0001.

**Figure 6 cells-13-01402-f006:**
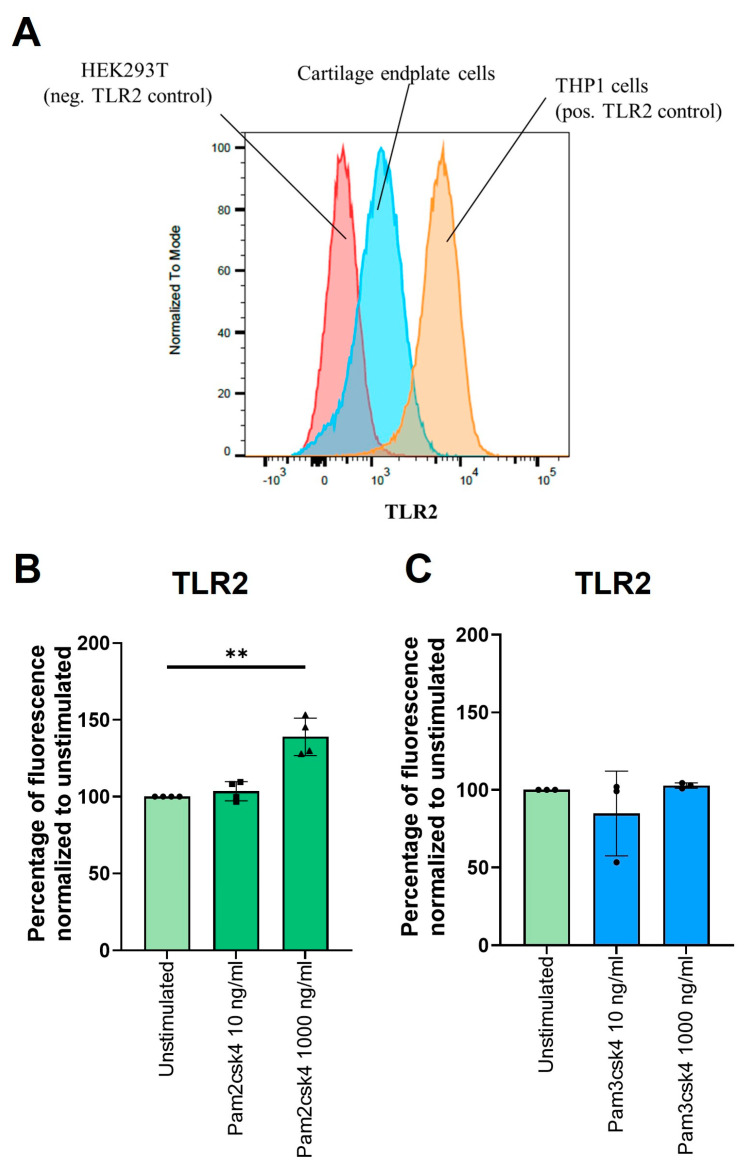
Protein expression of TLR2. (**A**) A representative case of TLR2 measured on CEPCs (blue), with HEK cells (red) serving as the negative control and THP1 cells (orange) serving as the positive control. (**B**,**C**) The effect of 72 h of stimulation with (**B**) Pam2csk4 and (**C**) Pam3csk4 at concentrations of 10 ng/mL and 1000 ng/mL on TLR2 levels illustrated as percentages normalized to unstimulated CEPCs. Significance was tested with one-way ANOVA on median fluorescence corrected for multiple comparisons using Dunnett’s multiple comparisons test. ** *p* < 0.01.

**Figure 7 cells-13-01402-f007:**
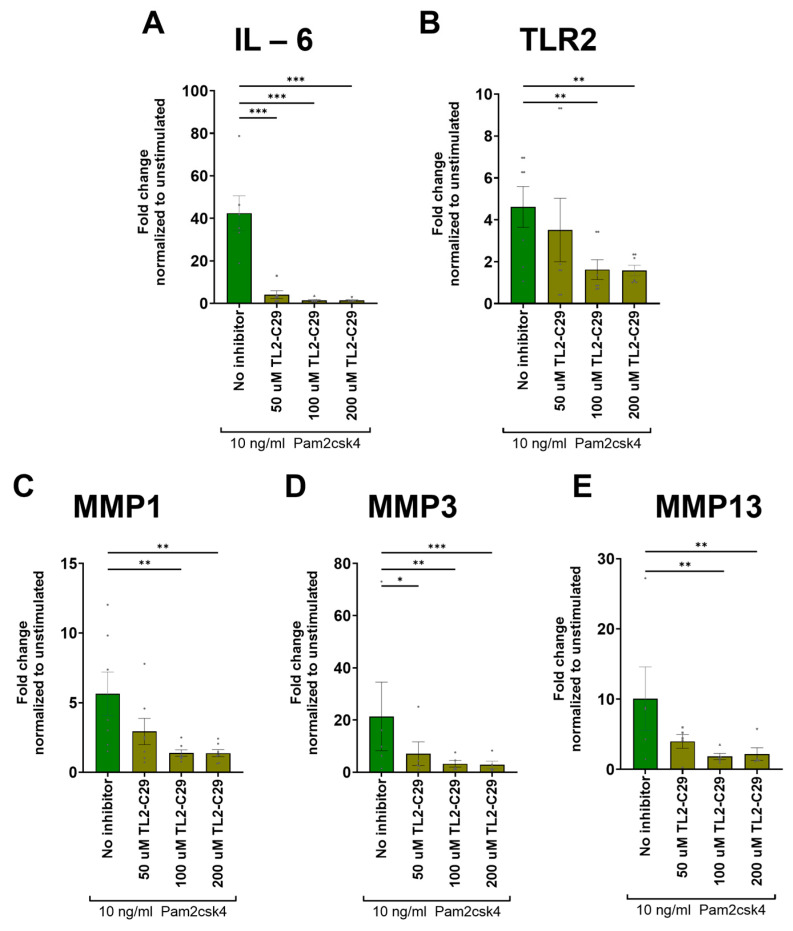
Inhibition of TLR2 signaling with three different dosages (50 µM, 100 µM, and 200 µM) of TL2-C29 added 2 h prior to adding Pam2csk4. Genes that showed upregulation through Pam2csk4 stimulation were used to test if this upregulation could be inhibited by blocking TLR2. The inflammatory gene (**A**) IL-6, the responding TLR (**B**) TLR2, as well as the proteases (**C**) MMP1, (**D**) MMP3 and (**E**) MMP13 were measured. The graphs indicate mean fold change values ± standard deviation. Statistical significance was tested on log_2_ fold change in DDC_T_ values by repeated measures one-way ANOVA, followed by multiple comparisons which compared each condition to the unstimulated condition at the respective timepoint. Dunnett’s statistical hypothesis testing was applied to correct for multiple comparisons. The asterisks signify significance: * *p* < 0.05, ** *p* < 0.001, *** *p* < 0.0001.

**Figure 8 cells-13-01402-f008:**
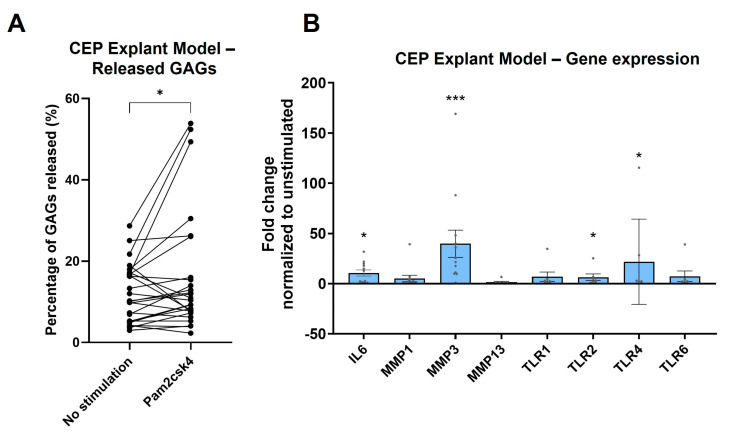
CEP explant model with half the CEP punch biopsy exposed to Pam2csk4. (**A**) The percentage of released glycosaminoglycans (GAGs) is illustrated, with each pair representing one endplate punch biopsy cut in half. Significance was tested using the Wilcoxon paired one-tailed *t*-test. (**B**) The transcriptome of CEP after 14 days of incubation with 10 µg/mL Pam2csk4 is illustrated as log_2_ fold change relative to the unstimulated half. Significance was tested using one-way ANOVA, corrected with the two-stage linear step-up procedure of Benjamini, Krieger, and Yekutieli. * *p* < 0.05, *** *p* < 0.001.

**Table 1 cells-13-01402-t001:** Primers used for qPCR.

Primers	Forward	Reverse
CCL2	5′-CAG CCA GAT GCA ATC AAT GCC-3′	5′-TGG AAT CCT GAA CCC ACT TCT-3′
GAPDH	5′-ATTCCACCCATGGCAAATTC-3′	5′-GGGATTTCCATTGATGACAAGC-3′
IL-6	5′-AGA CAG CCA CTC ACC TCT TCA G-3′	5′-TTC TGC CAG TGC CTC TTT GCT G-3′
IL-8	5′-GAG AGT GAT TGA GAG TGG ACC AC-3′	5′-CAC AAC CCT CTG CAC CCA GTT T-3′
MMP1	5′-ATG AAG CAG CCC AGA TGT GGA G-3′	5′-TGG TCC ACA TCT GCT CTT GGC A-3′
MMP3	5′-CAC TCA CAG ACC TGA CTC GGT T-3′	5′-AAG CAG GAT CAC AGT TGG CTG G-3′
MMP9	5′- GCCACTACTGTGCCTTTGAGTC-3′	5′-CCCTCAGAGAATCGCCAGTACT-3′
MMP13	5′-CCT TGA TGC CAT TAC CAG TCT CC-3′	5′-AAA CAG CTC CGC ATC AAC CTG C-3′
TLR1	5′-CAGTGTCTGGTACACGCATGGT-3′	5′-TTTCAAAAACCGTGTCTGTTAAGAGA-3′
TLR2	5′-GGCCAGCAAATTACCTGTGTG-3′	5′-AGGCGGACATCCTGAACCT-3′
TLR3	5′-CCTGGTTTGTTAATTGGATTAACGA-3′	5′-TGAGGTGGAGTGTTGCAAAGG-3′
TLR4	5′-CAGAGTTTCCTGCAATGGATCA-3′	5′-GCTTATCTGAAGGTGTTGCACAT-3′
TLR5	5′-TGCCTTGAAGCCTTCAGTTATG-3′	5′-CCAACCACCACCATGATGAG-3′
TLR6	5′-GAAGAAGAACAACCCTTTAGGATAGC-3′	5′-AGGCAAACAAAATGGAAGCTT-3′
TLR7	5′-TTTACCTGGATGGAAACCAGCTA-3′	5′-TCAAGGCTGAGAAGCTGTAAGCTA-3′
TLR8	5′-TTATGTGTTCCAGGAACTCAGAGAA-3′	5′-TAATACCCAAGTTGATAGTCGATAAGTTTG-3′
TLR9	5′-GGACCTCTGGTACTGCTTCCA-3′	5′-AAGCTCGTTGTACACCCAGTCT-3′
TLR10	5′-CTGATGACCAACTGCTCCAA-3′	5′-AGTCTGCGGGAACCTTTCTT-3′

## Data Availability

Data are provided within the manuscript or Appendix A.

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
