# Peer review of "The Expression of Toll-like Receptors in Cartilage Endplate Cells: A Role of Toll-like Receptor 2 in Pro-Inflammatory and Pro-Catabolic Gene Expression"

_cells, 2024, doi:10.3390/cells13171402_

Round 1

Reviewer 1 Report

Comments and Suggestions for Authors

The article is interesting and well-written.

The abstract is concise and well-explains the aims of the study, its methodology, the results and their implications. Similarly, the introducion clearly presents the state of the art regarding the gap in literature: the fact that the role of biological changes in the CEP has largely gone unexamined. The materials and methods section is complete, noteworthy is the fact that the approval of the ethical committee is clearly presented. The use of tables and figures enhances the readability of the article. results are clearly presented and their discussion is in line with them and coherent with the analysis of the literature.

Author Response

Comment 1: The abstract is concise and well-explains the aims of the study, its methodology, the results and their implications. Similarly, the introducion clearly presents the state of the art regarding the gap in literature: the fact that the role of biological changes in the CEP has largely gone unexamined. The materials and methods section is complete, noteworthy is the fact that the approval of the ethical committee is clearly presented. The use of tables and figures enhances the readability of the article. results are clearly presented and their discussion is in line with them and coherent with the analysis of the literature.

Reply 1: We thank the reviewer for their positive feedback to our manuscript.

Reviewer 2 Report

Comments and Suggestions for Authors

Cartilage endplate cells (CEPCs) are poorly characterized, hence this article provides a significant novel contribution to the common knowledge. It is well written and fully matches with the scope of cells.

introduction: modic change (MC) and MC1 could be better explained

cultures CEPC loose TLR7 and 8 expression: at which passage?

abstract: "enhanced TLR2 expression in MC1" - which figure does this show? Patients cohorts used for the different experimental settings (section 3.1) should be summarized in a table within the main body of the manuscript to get a better overview.

Could an age dependency be detected in regard to TLR expression?

line 54: comparison with nucleus pulposus cells - how about anulus fibrosus which is attached to the cartilaginous endplate?

line 72: intradiscal bacteria: how do they gain access to the disc?

DAPI: line 209, it stains only dead cells how were viable cells marked?

Fig. 1: why was this diagram type selected and not the same as in Fig. 2?

HEK and THP1: were 3 independent measurements undertaken with the controls? Please show SD.

Immunohistochemistry: was it semiquantified?

Figure 2B: I would indicate that the difference is significant.

Line 358/384: surplus point

Fig. 6: why were the proteins analysed after 72 hours

Line 474: TNF/IL-1 TLR independent regulation- perhaps it should be explained earlier

Line 496: CEP resorption – overinterpretation - only GAG release was measured, no information about collagen network destruction

Author Response

Please see attachment for a full point-by-point response.

Reviewer 3 Report

Comments and Suggestions for Authors

Congratulations for the paper. My comments are:

·       The abstract must have the subsections well defined. This helps with the compression of the text.

·       At the end of the introduction, the objective should be better written.

·       Table 1 needs to be better designed

·       En results section, the p value should have only two decimal places, if p>0.05

·       The discussion should begin with "The main finding of this study..."

·       Authors should include a limitations section at the end of the discussion section.

·       The style of reference is not correct.

Comments on the Quality of English Language

the text must be reviewed by a native English speaker.

Author Response

Please see attachment for a point-by-point response.

Round 2

Reviewer 2 Report

Comments and Suggestions for Authors

The authors fully addressed my previous comments.